# Efficient encoding of spectrotemporal information for bat echolocation

**Adarsh Chitradurga Achutha**[1], **Herbert Peremans**[2], **Uwe Firzlaff**[3], **Dieter Vanderelst**[4]*

**1** Mechanical and Materials Engineering, University of Cincinnati, Cincinnati, Ohio, United States of America, **2** Department of Engineering Management, University of Antwerp, Antwerp, Belgium, **3** Chair of Zoology, School of Life Sciences, Technical University of Munich, Freising, Germany, **4** Department of Biological Sciences, University of Cincinnati, Cincinnati, Ohio, United States of America

* vanderdt@ucmail.uc.edu

**Data Availability Statement:** All data and code are available at https://doi.org/10.6084/m9.figshare.14573652.v1.

## Abstract

In most animals, natural stimuli are characterized by a high degree of redundancy, limiting the ensemble of ecologically valid stimuli to a significantly reduced subspace of the representation space. Neural encodings can exploit this redundancy and increase sensing efficiency by generating low-dimensional representations that retain all information essential to support behavior. In this study, we investigate whether such an efficient encoding can be found to support a broad range of echolocation tasks in bats. Starting from an ensemble of echo signals collected with a biomimetic sonar system in natural indoor and outdoor environments, we use independent component analysis to derive a low-dimensional encoding of the output of a cochlear model. We show that this compressive encoding retains all essential information. To this end, we simulate a range of psycho-acoustic experiments with bats. In these simulations, we train a set of neural networks to use the encoded echoes as input while performing the experiments. The results show that the neural networks' performance is at least as good as that of the bats. We conclude that our results indicate that efficient encoding of echo information is feasible and, given its many advantages, very likely to be employed by bats. Previous studies have demonstrated that low-dimensional encodings allow for task resolution at a relatively high level. In contrast to previous work in this area, we show that high performance can also be achieved when low-dimensional filters are derived from a data set of realistic echo signals, not tailored to specific experimental conditions.

## Author summary

We show that complex (and simple) echoes from real environments can be efficiently and effectively represented using a small set of filters. Critically, we show that high performance across a range of tasks can be achieved when low-dimensional filters are derived from a data set of realistic echo signals, not tailored to specific experimental conditions. The redundancy in echoic information opens up the opportunity for efficient encoding, reducing the computational load of echo processing as well as the memory load for storing the information. Therefore, we predict the auditory system of bats to capitalize on this

**Funding:** The author(s) received no specific funding for this work.

**Competing interests:** The authors have declared that no competing interests exist.

opportunity for efficient coding by implementing filters with spectrotemporal properties akin to those hypothesized here. Indeed, the filters we obtain here are similar to those found in other animals and other sensing capabilities. Our results indicate that bats could exploit the redundancy in sonar signals to implement an efficient neural encoding of the relevant information.

## Introduction

Many natural stimuli encountered by animals are characterized by a high degree of redundancy [1]. Efficient neural encoding retains essential information while reducing this redundancy. By extracting the most crucial aspects of stimuli, the efficiency of sensing is drastically increased [2]. For echolocating bats, the time of arrival of echoes, which conveys the target's distance, is the most relevant sensory information [3]. In addition, the spectral content and intensity of the echoes also convey essential information for localizing and recognizing targets [4, 5].

Earlier results from robotic sonar suggest that echo signals, like many other natural stimuli, are highly redundant. For example, [6] presented a bio-mimetic system that could differentiate the head and tail sides of a coin. This binaural system reduced the 2.4 ms long, 60 kHz waveform at each receiver to a 16-value vector. This corresponded to 0.15 samples per millisecond instead of the 120 samples prescribed by the Nyquist criterion. Collecting many broadband echoes (100–30 Khz) in different natural bat habitats [7] demonstrated successful place and pose recognition based on these echoes. Each echo was represented using a distance-intensity profile of fewer than 100 samples, or less than 3 samples per millisecond. References [8] and [9] presented simulations and a robotic model of obstacle avoidance in bats. Their obstacle avoidance strategy only used the (sign of the) interaural level difference to steer the artificial bats around obstacles, demonstrating successful albeit simple sensorimotor control using a highly reduced representation of the echo train.

The previous work, referred to above, showed the possibility to compress echoic information in the context of a particular task, e.g., scene recognition [7] or object recognition [6, 10]. However, to be truly useful, efficient encoding must support many different echolocation based behaviors. Hence, to maximally exploit this apparent redundancy in echo signals, it can be presumed that bats have evolved efficient and task-independent neural encoding strategies for extracting relevant echo features. While specific neural encodings in various areas of the bat's brain have been studied in great detail, to the best of our knowledge encodings that explicitly take into account the redundancy in echo signals have not been studied systematically yet. Here, we set out to devise such an efficient encoding scheme for echo signals.

First, we collect a large sample of representative stimuli, i.c., echoes from different indoor and outdoor environments, and convert them to cochleograms, as proposed by Lutz Wiegrebe, to whose memory we dedicate this paper [11]. Next, we employ Independent Component Analysis to derive a set of filters that most efficiently encode the ensemble of cochleograms. This same approach has been used to good effect in other sensory domains, including visual [12] and auditory [13] perception. The resulting filters can be interpreted as the spectrotemporal receptive fields of a set of hypothetical neurons [14]. We find that 25 filters can be used to encode 99% of the variance in the cochleograms derived from echoes collected in various environments. Next, we present simulations showing that these filters retain sufficient information to complete several sonar-based tasks on which bats have been tested before. In particular, we show that the filters allow for accurate object recognition, monaural target localization

(range and elevation), and scene recognition. Crucially, we show that high performance across a range of tasks can be achieved when low-dimensional filters are derived from echo signals not tailored to specific sonar-based tasks.

These results show that the (lossy) compression performed by the filters provides a substantial reduction in the amount of echoic information that needs to be encoded, processed, and stored while retaining the crucial aspects of the stimuli. The proposed filters have been derived on theoretical grounds, but, as discussed, evidence from auditory and visual encoding in both bats and other animals supports their biological plausibility.

## Methods

### Overview

The overall approach of the current study is as follows (also depicted in Fig 1). We start by collecting echoes ($N$ = 1014) in 21 natural bat habitats and indoor environments. While not intended to be exhaustive, this dataset covers a broad range of environments containing human-made and natural reflectors of varying complexity that bats might conceivably encounter. Using a functional model of the auditory periphery of the bat [11], we convert the echoes to cochleograms. The information in the cochleogram is a good approximation of that contained in the neural activity at the auditory nerve [15]. Next, based on this database of ecologically relevant cochleograms, we derive a set of filters for encoding the cochleogram using the technique of Independent Component Analysis [16].

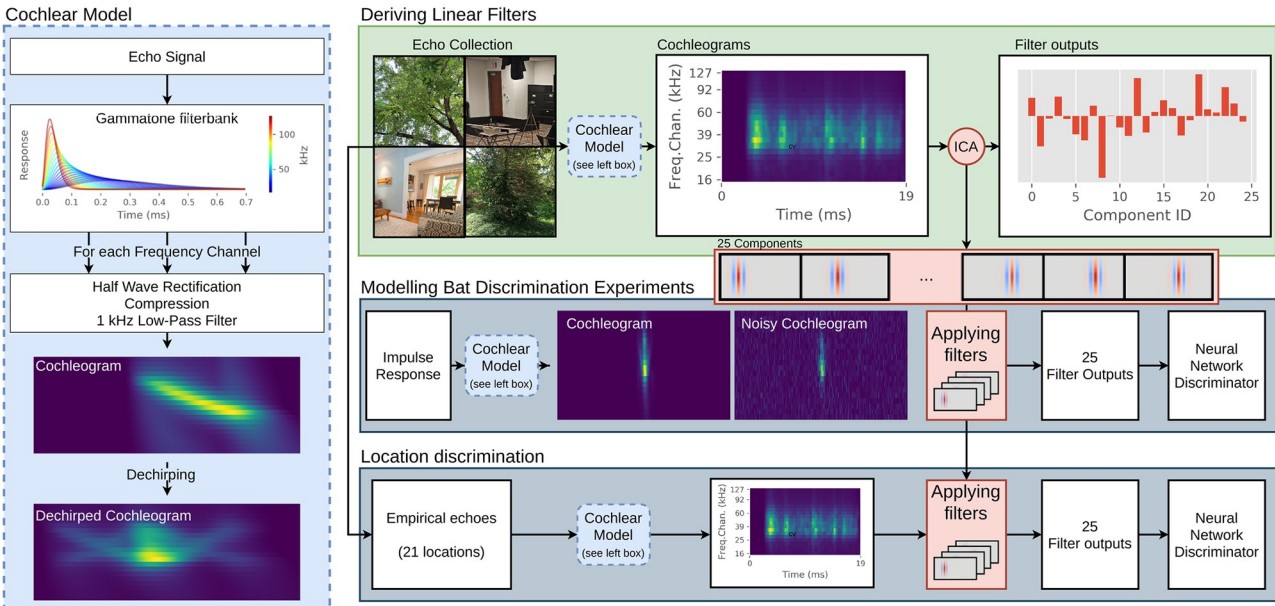

**Fig 1. General outline of the approach of this paper. Top** Echoes collected in various environments are converted to cochleograms using a model of the auditory periphery [11]. From these cochleograms, we derive a set of 25 independent components (=filters) using Independent Component Analysis (ICA). Using these components as a basis for cochleogram encoding allows compressing each cochleogram into a set of 25 values (=filter outputs). **Middle** To test whether these 25 filters retain sufficient information to explain the behavior of bats in a range of psychophysical tasks, we model four experiments. For each of these experiments, we generate phantom echoes based on the impulse responses as used in the bat experiments. Next, we convert these phantom echoes to cochleograms and add internal noise. These cochleograms are then encoded with the 25 filters derived from the echo database. Finally, a neural network is trained to assess whether this encoding retains sufficient information to achieve a similar discrimination performance as the bats. **Bottom** In addition to modeling four bat experiments, we also test whether the encoding retains enough information to memorize and recognize the 21 locations at which the data were collected.

To verify that this encoding retains the relevant spectrotemporal echo information useful to a bat, we simulate several previously published behavioral discrimination experiments with bats. In our experiments, we generate echoes similar to those used in the bat experiments and encode the associated cochleograms with the proposed efficient encoding scheme. We then use neural networks to assess whether the information retained by this compressive encoding is sufficient to attain a performance level comparable to that of the bats in the discriminations experiments. See [17] for a similar approach. As a final test, we also assess whether the information retained by the encoding is sufficient to memorize and discriminate between the cochleograms collected at the 21 different locations, a capability required for place, and scene recognition by bats [7].

## Deriving an efficient cochlear encoding

We collected echo data using a sonar data acquisition device mounted on a tripod (Fig B in S1 Text). The device consisted of a Sensecomp 7000 broadband emitter. A 1-millisecond FM-pulse sweeping down from 70 kHz to 30 kHz (hyperbolic sweep) was used. This band corresponds to the lower frequency range used by many species of bats in echolocation [18]. The device featured two Knowles microphones. However, in this study, only one of the microphones was used. Echoes recorded by the microphone were sampled at 360 kHz.

Echoes were collected in several outdoor and indoor environments. Outdoors, we ensonified hedgerows, dense vegetation, plants, tree foliage, and shrubs. Indoors, data were collected in various rooms of a private residence, in lab spaces at the University of Cincinnati, and inside a barn. We selected these locations to include simple (human-made) reflectors and highly complex stochastic reflectors (e.g., leafy foliage). In total, we collected 1014 echoes, 500 of which were collected indoors. Echoes were collected in batches of 30 to 50 at 21 different locations. Some examples of the locations we used are shown in Fig 1. The echoes had a duration of 19 ms, corresponding with a maximal range of 3.2 m (=$343 m/s \times 0.019/2$) for a speed of sound of 343m/sec. The duration of 19 ms was dictated by the size of the onboard memory of the data acquisition device. Stilz and Schnitzler [19] found, that depending on atmospheric conditions, echolocation frequency, and the dynamic range of the sonar system, the maximum range for extended backgrounds such as a forest edge can be as short as 2.4m. Therefore, we propose that the chosen echo length while at the lower end falls within the ecologically relevant range.

While collecting data, the position and orientation of the ensonification device were changed between subsequent emissions. We moved the device pseudo-randomly through each space by displacing it by about 20 centimeters between measurements and turning it up to 90 degrees. For each position and orientation of the device, three measurements were taken in succession, separated by 1 second.

The echoes, for each of the three repeats, were converted into cochleograms using the functional model of the middle and inner ear processing in the bat, as proposed by Wiegrebe [11] (see Fig 1). We averaged across the cochleograms for the three repeats to increase the signal-to-noise ratio. The middle and inner ear processing model consists of a bank of gammatone filters followed by half-wave rectification and exponential compression. Finally, each frequency channel's output is low pass filtered with a cut-off frequency of 1 kHz. As the bat has knowledge of its emission and as we ignore possible Doppler-shifts (hyperbolic FM sweeps are maximally Doppler-shift resilient [20, 21]), the frequency modulation present in each subecho the cochleograms consist of can be compensated for by a 'dechirping' operation. Through this 'dechirping' mechanism, we shift the response in each cochlear frequency channel in time to align the responses (see Fig 1). A similar compensation mechanism is included in both the

SCAT model proposed by Saillant et al. [22] and the model proposed by Wiegrebe [11], implemented through autocorrelation. In the current study, the 20 center frequencies for the gammatone filter bank were spaced by Equivalent Rectangular Bandwidths [23]. An example of a cochleogram is shown in Fig 1. The center frequencies are listed in S1 Text.

Next, each cochleogram $S_j$, $j = 1, \cdots, N$ is converted into a vector $x_j$ by concatenating the columns of the cochleogram. The efficient encoding we propose assumes that this observed vector $x_j$ can be written as a linear mixture of basic components,

$$\mathbf{x_j} = \sum_{i=1}^{N} c_{j,i} \cdot \mathbf{\Psi_i} = \mathbf{A} \cdot \mathbf{c_j} \tag{1}$$

with the basic components $\mathbf{\Psi_i}$, $i = 1, \cdots, N$ making up the columns of the matrix $\mathbf{A} = [\mathbf{\Psi_1} \, \mathbf{\Psi_2} \cdots \mathbf{\Psi_N}]$ and $\mathbf{c_j}$ a vector of statistically independent weights with the $i$-th component of this vector denoted by $c_{j,i}$. Given the dataset of cochleograms, the ICA technique will determine the matrix $\mathbf{A}$ that minimizes the multi-information, a generalization of mutual information measuring the statistical dependence between multiple variables, of the weights $\mathbf{c_j}$. By inverting the concatenating operation performed on the cochleograms, we can interpret the basic components $\mathbf{\Psi_i}$, $i = 1, \cdots, N$, just like the cochleograms, as functions of time $t$ and frequency $f$. In this paper, we use the FastICA [24] algorithm as implemented by the scikit-learn Python package [25] to derive the basic components from the set of collected cochleograms.

In principle, the set of weights $\mathbf{c_j}$ encodes the cochleograms without loss, i.e., the dimensions of the vectors $x_j$ and $c_j$ are the same. However, in this paper, our goal is to assess whether a reduced set of basic components $\mathbf{\Psi_i}$, $i = 1, \cdots, M$ with $M \ll N$ can capture sufficient spectrotemporal information to successfully support a broad range of typical echolocation tasks. Hence, before the ICA proper is applied, the cochleograms first undergo a preprocessing step consisting of the removal of the mean cochleogram and principal component analysis (PCA). Projecting the cochleograms onto their principal components removes linear correlations and allows, by dropping dimensions with low variance, a dimensional reduction. The PCA performed on the cochleogram dataset showed that 25 components could capture over 99% of the variance in the cochleograms. Next, after mapping the cochleograms onto this reduced 25-dimensional Principal Components space, the 25 independent components representing the data best are determined. As this preprocessing step is included in the FastICA implementation, we refer to both these processing steps as ICA in Fig 1.

An example of a cochleogram $S_j$ converted to its 25 dimensional representation $c_j$ is given in Fig 2.

## Modeling behavioral data

To demonstrate that this compressive encoding retains sufficient information to explain bats' behavior in various experiments, we modeled four previously published behavioral experiments. In modeling each of these, we employed the same approach outlined in Fig 1.

We generated artificial echoes according to the same procedure used in each behavioral experiment. Three out of four of the behavioral experiments considered used a phantom target paradigm. In these experiments, the bat's emission was recorded using a microphone and convolved in real-time with a target impulse response. The result was played back to the bat. We generated impulse responses mimicking those used in the experiments and convolved them with the same emission signal used to collect the real sonar echoes, i.e., a 1-millisecond FM-pulse sweeping down from 70 to 30 kHz. The fourth experiment used real targets (i.e., small beads). We approximated those as reflecting a simple copy of the incident emission.

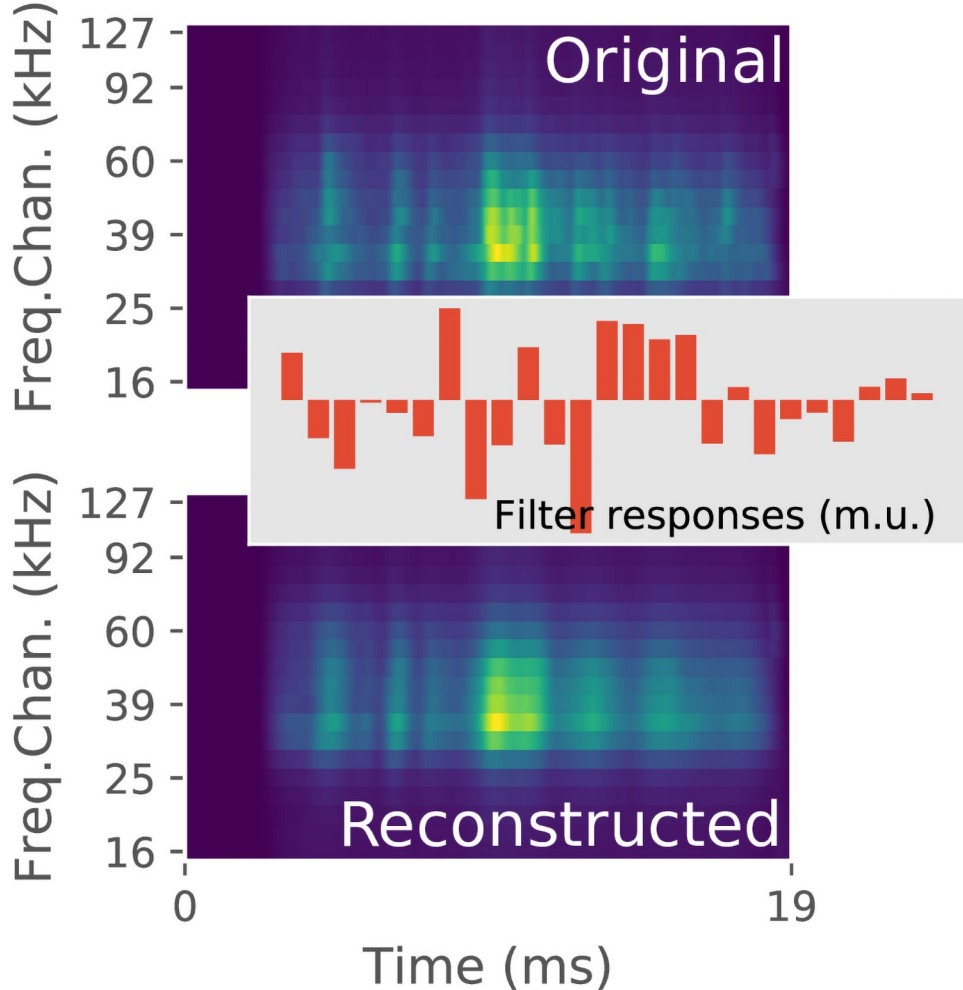

**Fig 2. Example of a cochleogram from the ensonification data.** Its 25D representation is shown in the center of the figure. At the bottom the reconstructed cochleogram (from the 25D represention) is shown. The reconstructed version is a smoothed version of the original. (m.u.: model units).

These artificial echoes were converted to cochleograms employing the same model [11] used to process the ensonification data. Internal noise (see below for details) was added to the cochleograms before encoding them using the 25 independent components derived from the echo database. Next, we ascertained that this encoding retained essential spectrotemporal information. We did this by training neural networks on the compressed encoding of the cochleograms to determine discrimination thresholds that can be compared to the behavioral findings reported in the earlier studies.

For each experiment modeled, we constructed a separate training and testing data set. The training set was always used exclusively to train the neural network, whereas all performances reported here are exclusively based on the separate test set. For some experiments, we wanted to test whether the neural network could generalize from the training set. In this case, the training and the test set have been generated with different parameter settings. We will discuss this where appropriate.

The networks consisted of a 25 node input layer matching the dimension of the 25 independent component space. We used two hidden layers, each with 50 nodes. The number of nodes

was selected for computational convenience, and we did not attempt to optimize the networks' size. The hidden layer neurons were Rectified Linear Units. The networks had one output node, with a sigmoid activation function. The experiments we modeled used two-alternative forced-choice (2AFC) tasks. Therefore, we trained the networks to generate an output = 1 for inputs corresponding with a rewarded stimulus and output = 0 for inputs corresponding with a non-rewarded stimulus. The loss function was the absolute difference between the desired and actual activation of the output node. The training was done using RMSProp (Root Mean Square Propagation), as implemented in Keras [26].

**Noise model.**   We added Gaussian (zero mean) noise to the cochleograms before applying the compressive encoding to model the effects of internal noise on the bat's decision process [11, 27]. To derive the variance of the noise, we employed the calibration procedure proposed by [27], i.e., we established the noise level that resulted in similar echo intensity discrimination performance as reported for bats. Reference [28] summarizes intensity discrimination threshold experiments, citing Just-Noticeable Difference (JND) values for echo intensity ranging from 1 to 5 dB. We chose to use a value at the upper end of that range, i.e., 5 dB, as this same value (for 70% correct decisions) was reported in phantom target experiments similar to the ones we will be simulating [29].

In particular, we generated a cochleogram $S_{ref}$ corresponding with a reference echo amplitude. A second cochleogram $S_{+5}$ was generated with an echo amplitude 5 dB higher than the reference amplitude. Next, we iteratively searched for the level of Gaussian noise, $\mathcal{N}(0, \sigma)$, that allowed telling $S_{+5}$ apart from $S_{ref}$ for 70% of the noise realizations. The value of $\sigma$ that allowed for 70% correct decisions was approximately 0.1 (S1 Text). A cochleogram with added noise is shown in Fig 1. Using $\sigma = 0.1$ gives the cochleograms in this paper a dynamic range of about 42 dB, i.e., the maximum value across all cochleograms is approximately 15.

In the remainder of this paper, all simulated echoes were generated with the same reference amplitude. In those cases where amplitudes of echoes were varied, the amplitude roving was done around the reference value.

**Experiment 1: Encoding temporal information.**   To assess whether the compressive encoding retained sufficient temporal information, we mimicked the experiments that quantified the just noticeable difference in echo delay in bats (see [30] for references). For example, in the experiments described by Denzinger and Schnitzler [31], the bats were rewarded for discriminating a phantom target echo at a fixed delay from echoes with a longer and variable delay.

We modeled this absolute delay discrimination experiment by generating target impulse responses consisting of a single impulse. The rewarded target impulse response was fixed at a delay of 11 ms. Unrewarded target impulse responses were shifted backward from 1000 $\mu$s to 50 $\mu$s relative to the fixed target. We applied amplitude roving by randomly varying the echo amplitudes over a 30 dB range to exclude overall echo-level cues. The resulting echoes were converted to cochleograms, noise was added, and the result was encoded with the 25 filters derived above.

**Experiment 2: Encoding simple reflector descriptions.**   To assess whether the compressive encoding retained sufficient information about simple target impulse responses, we investigated the discrimination of stimuli containing two echoes separated by varying time delays, as proposed in [32]. In these experiments, *Megaderma lyra* were presented with phantom targets defined by an impulse response consisting of two impulses separated by a variable time delay ranging from 1.3 to 26 $\mu$s. In some of the experimental conditions, these two echoes had unequal strength. These short time delays, falling within the cochlear frequency channels' integration time, result in a notch in the phantom targets' spectral image. Indeed, assuming the received echo $x(t)$ consists of two delayed copies of the call $e(t)$ with the second one possibly

amplified/attenuated with respect to the first one,

$$x(t) = e(t) + ae(t - \tau),\tag{2}$$

the spectrum of such an echo can be written as

$$X(f) = E(f) \cdot (1 + a \exp^{-j2\pi f\tau}),\tag{3}$$

with $E(f)$ the spectrum of the emission. This spectrum contains notches at frequencies $f^-$, assuming no phase shift between echoes, that depend on the time delay $\tau$

$$f^- = \frac{(2m + 1)}{2}\frac{1}{\tau},\tag{4}$$

with $m$ an integer. The depth of the notch depends on the ratio $a$ of the leading and trailing echo amplitudes with maximum depth achieved for $a = 1$. In the original experiments, bats were trained to discriminate a stimulus with a reference time delay 7.77 $\mu$s (or reference notch frequency of 64.4 kHz) from stimuli with different time delays (or notch frequencies).

In keeping with the study by Schmidt [32], we generated stimuli consisting of two echoes with the delay $\tau$ between them varied such that the corresponding notch falls in the interval from 16 to 70 kHz (2 kHz steps), corresponding with the passband of our sonar system. As in the most challenging of the original study's experimental conditions, the leading echo's amplitude was set 6 dB lower than the trailing echo's, resulting in less pronounced notches. Similar to the previous experiment, we applied amplitude roving by randomly varying the two echoes' amplitudes over a 30 dB range. The delayed echoes were again converted to cochleograms, noise was added, and the results were mapped onto the same 25 independent components (or filters).

Note that irrespective of whether the spectral image is used directly by the bats to solve this task, as concluded by Schmidt [32], or whether this spectral image is transformed into a time-domain representation of the target impulse response first, as suggested by in references [22, 33], the loss of spectral information will harm discrimination performance in this experiment. Hence, while the previous experiment studied how the proposed compressive encoding preserves temporal information, this experiment tests how well spectral information is preserved.

**Experiment 3: Encoding scale-invariant reflector descriptions.**   To assess whether the compressive encoding retained sufficient information about more complex target impulse responses as well, we mimicked an experiment on scale-invariant reflector recognition. In this experiment, Firzlaff et al. [34] trained bats to discriminate echoes resulting from two different impulse responses consisting of 12 impulses each. After the bats had been trained to distinguish between these two successfully, they presented the bats with scaled versions of the target impulse responses mimicking decreased or increased reflector sizes. Scaling entailed multiplying the impulse amplitudes with the square of the scale factor (area of target scales as the square of linear scale factor) and compressing or expanding the target impulse responses along the time axis with the scale factor. As the bats could still discriminate between scaled versions of the original target impulse responses, the experiments showed that the bats could generalize from the trained target impulse responses to their scaled versions.

Similar to the approach taken by Firzlaff et al. [34], we generated two impulse responses consisting of 12 impulses randomly distributed over a 1.86 ms time interval. Scaled versions of these two impulse responses were generated using the same scaling procedure and the same range of scale factors applied in the bat experiments, i.e., a scale factor ranging from 0.65 to 1.5. We applied 15 different scalings in this range. Again, the impulse responses were converted to noisy cochleograms before encoding them using the same 25 filters. To mimic the

original study's experimental design, we trained the neural network to discriminate the impulse responses at scale 1. Next, we tested the neural network's ability to generalize this discrimination to the scaled versions.

**Experiment 4: Encoding spectral information.**   To assess whether the compressive encoding retained sufficient spectral information for spatial localization of targets, we modeled the elevation discrimination experiments reported in references [5, 35]. As in other mammals, the pinnae of bats generate acoustic cues that aid the localization of echo-producing reflectors. Experiments have shown that the spectral cues imposed by the pinnae are particularly important for localization in the vertical plane, for example, [5, 35–37].

To the best of our knowledge, no phantom target experiments have been used to test bat's elevation discrimination performance. Therefore, we cannot exactly duplicate a particular experimental setup. Instead, we generated cochleograms using the head-related transfer function (HRTF) of *Phillostomus discolor* [38, 39] of echoes from virtual targets at azimuth = 0 degrees and varying elevation angle between -20 degrees and 20 degrees. This interval represents the interval of best elevation discrimination found by Lawrence, Simmons and Wotton [5, 35]. We trained the neural network to return a 0 when the 25D input vector was derived from echoes filtered with the HRTF for negative elevations and a 1 for echoes filtered with the HRTF for positive elevations. While this setup differs from the bat experiments referred to above to determine the vertical angular acuity, we propose that it also provides an estimate of the vertical angular acuity. Furthermore, even if the performances in the real and simulated experiments are not directly comparable, solving this discrimination task for target positions close to the horizontal plane will require the neural network to distinguish subtle location-dependent HRTF cues showing that the compressive encoding retains those cues.

## Memorizing acoustic signatures

In previous work, we established that cochleograms contain sufficient information to recognize scenes and sonar poses (location and orientation) [7]. In this experiment, we assessed whether the measured cochleograms would still allow place recognition after being projected into the compressed independent components space. To test this, we trained a neural network to associate each of the 25D-vector encodings (see Fig 1) of the empirically collected echoes with the location at which they were collected.

The network used for memorizing acoustic signatures differed from those used in modeling the 2AFC behavioral experiments. This network had three hidden layers with 50, 100, and 50 Rectified Linear Units. The output layer had 21 units, each corresponding to one of the 21 locations where acoustic data was collected. The output layer had a softmax activation function. This activation function normalizes the neurons' output into a probability distribution. Therefore, the output of this network could be interpreted as a probability distribution across the 21 locations. Categorical Cross Entropy was used as a loss function, and optimization was performed using the Adaptive Moment Estimation (adam) algorithm [40].

## Results

### Basic components

Fig 3A–3D visualizes four of the basic components $\Psi_i$, $i = 1, \cdots, 25$, derived from the echo database. As is clear from this figure, the components show Gabor-like properties along the time dimension. Each component is sensitive to a given delay (or distance) and shows some suppression for shorter and longer delays. Note that because of the 'dechirping' operation performed before calculating the cochleograms, all the filters' Gabor-like responses are vertically oriented.

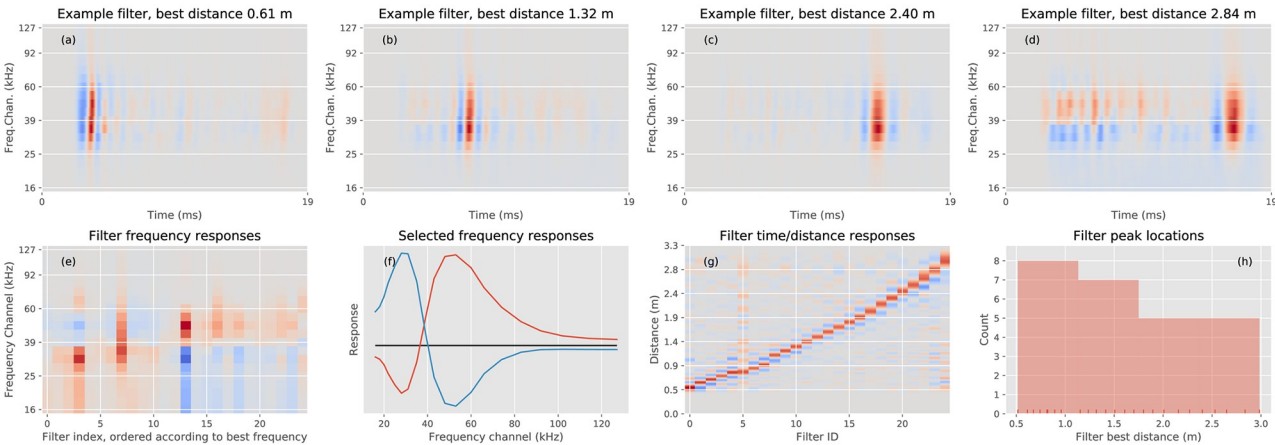

**Fig 3. The basic components and their properties. a-d** Selected examples of independent components. **e** The frequency response of the 25 components (ordered according to most sensitive frequency). For each component, its time-average as a function of frequency is plotted. **f** The frequency response of two selected components (i.e., two columns from panel **e**). **g** The temporal response of the 25 components (ordered according to most sensitive delay or distance (=343 × *delay*/2), with speed of sound 343m/sec). For each component, its frequency-average as a function of distance (time) is plotted. **h** Histogram of the best distances (times) of the 25 components.

To investigate the spectral properties of this set of components, we calculate for each component and each frequency channel the mean value along the time dimension (see Fig 3E), and likewise, for the temporal properties, we calculate for each component and for each time the mean value along the frequency dimension (see Fig 3G). As can be seen from Fig 3E, the frequency responses of the components are centered on the range of frequencies most salient in the echoes, i.e., a band around 35 kHz to which the sonar is most sensitive. However, individual components differ somewhat in the frequency to which they are most sensitive. Some components exhibit a similar center-surround characteristic along the frequency axis, as is apparent along the time axis. They are most (least) sensitive to specific frequencies (with some having multiple peaks in their frequency response) and have a region of inactivation (activation) above or below this range, as illustrated by the time-average of two selected components shown in Fig 3F. These two components are most sensitive to different frequencies and have a suppressed frequency region as well. Note that these differences in frequency response allow the set of independent components to encode the spectral properties of the cochleograms.

Fig 3G shows that the components also differ in the delay or distance to which they are most sensitive. This figure shows that the 25 components each have a slightly different best time or distance response. There is a tendency for the components to be most sensitive to shorter time delays. A histogram (Fig 3H) of the best distances (delays) of the 25 components confirms this. Also, the components respond at a faster time-scale at shorter distances and tend to become slower at longer distances, as can be seen in Fig 3A–3D. This can be understood by noting that absorption in air is higher for higher frequencies. Hence, high-frequency contributions to real echoes will be more pronounced at shorter distances and become less so as the distance increases. These high-frequency contributions will stimulate the high-frequency channels of the cochleogram, and because of the larger bandwidths of these channels, this will result in a faster overall cochlear response. Indeed, from the Gammatone filterbank responses shown in Fig 1 (red = high frequency channels, blue = low frequency channels), we note that the low-frequency channels have a much slower response time than the high-frequency channels, see [11]. This indicates that the independent components we derived from

the real echoes have correctly captured the physical constraints shaping the spectrotemporal cues present in the cochleograms.

The derived components encode the spectrotemporal information in the cochleograms by being most sensitive to different time delays and frequencies. The filters exhibit evident center-surround characteristics along the time axis: they have a best delay surrounded by suppression regions. Similar, though somewhat less pronounced, features also emerge along the frequency axis.

## Modeling behavioral experiments

Fig 4 shows the results from modeling the four behavioral tasks. As can be ascertained from Fig 4A, when mimicking echo delay discrimination experiments by feeding the compressed encodings into a neural network, the just-noticeable difference (JND) in echo delay is about 12 $\mu$s.

The JND in echo delay of the model is smaller than the JND values observed in the literature. However, it should be noted that the range of values reported for JND in echo delay is substantial and depends on the specific experimental conditions. Goerlitz et al. [30] quote a range of 36 to 167 $\mu$s, across experiments. In their experiments, Goerlitz et al. [30] found values of about 20 to 25 $\mu$s for the bat *Glossophaga soricina*. The neural network's performance shows that the compressive encoding retains sufficient temporal information to perform temporal discriminations, at least as good as the bats'.

Fig 4B shows that the neural network was also able to discriminate between different notch locations (or time delays between two impulses) when mimicking the experiments of Schmidt [32]. In particular, the network's performance was closest to the experimental condition in which the bats scored best, i.e., when the trailing and leading echo amplitudes were the same. However, in our simulations, these amplitudes differed by 6 dB. The bats' performance for this, more challenging, experimental condition is also plotted in Fig 4B. Note that the notch's reference location (=rewarded stimulus) falls in the center of the blue shaded region, which shows the mean spectrum of all echoes from the database. To make this happen, we shifted the corresponding rewarded delay from 7.77 $\mu$s in the bat experiment to 10 $\mu$s in the simulation.

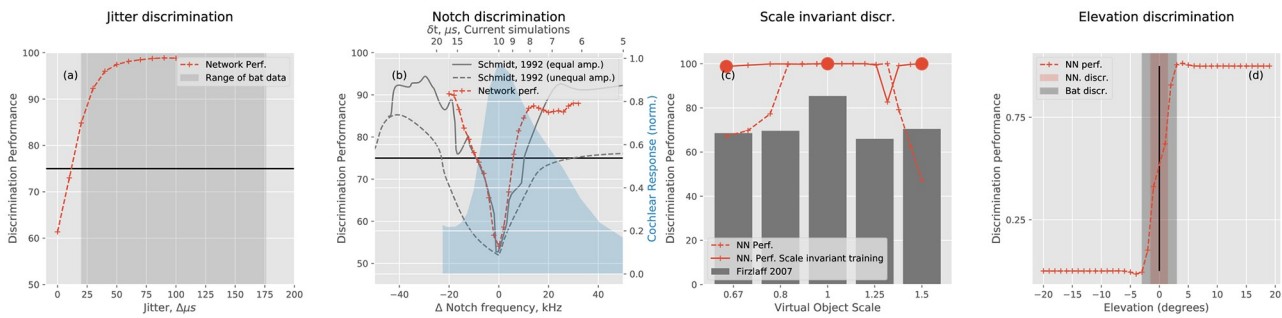

**Fig 4. All red curves depict neural network performances.** Grey data shows behavioral data taken from previously published experiments. **(a)** The results from modeling a delay discrimination task, e.g., as described by Denzinger and Schnitzler [31]. In gray, we depict the range of discrimination thresholds found in the literature ([30] and references therein). **(b)** Results from mimicking the frequency notch (or internal delay) discrimination experiments by Schmidt [32]. This panel shows the behavioral results of two experimental conditions tested. In one condition, the leading and the trailing echo had equal amplitudes. The amplitudes differed by 6 dB in a second condition, making discrimination harder for the bats (see text). **(c)** Firzlaff et al. [34] trained bats to discriminate two phantom objects (scale 1). Next, the bats were able to generalize and discriminate between scaled versions of these objects. The neural network was able to make the same generalization but for scale = 1.5. A second network trained on a subset of scaled phantom objects (red dots) could better interpolate to intermediate scales. **(d)** Results from training a network to discriminate between phantom echoes coming from above and below the horizon. This results in an estimate of the vertical angle acuity that is better but comparable to the value measured in the behavioral experiments conducted by Lawrence, Simmons and Wotton [5, 35].

This was done to ensure that the notch fell within the sonar system's frequency range for the entire interval of tested delays. The neural network's discrimination curve shows that encoding the cochleogram using only 25 components does not hamper the successful completion of this discrimination task. Indeed, the performance of the neural network is similar but uniformly better than that of the bats.

From the results shown in Fig 4C, we conclude that similar to the bats in the scale-invariant object recognition experiments described in [34], the neural network learned to discriminate the two (complex) target impulse responses at scale 1 reliably. This discrimination capability seems to generalize to scaled versions of the same target impulse responses, except for scale = 1.5. Only at the largest scale, the network's performance is notably lower than that observed in bats.

The variability in performance as a function of the scale indicates that the independent components we derived are not scale-invariant representations. However, when cochleograms (and derived representations) are not scale invariant, for bats to be nevertheless able to perform scale-invariant object recognition, the relevant information should still be contained in these representations. To demonstrate that further processing could still extract scale-invariant reflector information from our compressive encoding, we trained a second neural network to discriminate between target impulse responses scaled with factors 0.65, 1, and 1.5. Note that the only difference with the previous simulation is the presentation of a broader range of examples during the neural network's learning phase. As is clear from the results shown in Fig 4C, the network trained on these three scaled variants of the original target impulse responses can generalize its discrimination capability to other intermediate scales. Hence, this suggests that the encoded cochleograms do indeed retain sufficient information to derive a scale-invariant representation. How bats might accomplish this is beyond the scope of this paper.

Finally, the encoded cochleograms also retained the spectral information required to perform elevation discrimination. As shown in Fig 4D, the neural network was able to discriminate (75% criterion) angles as little as 1–2 degrees above or below the horizon. This is somewhat better but corresponds well with the 3-degree discrimination threshold observed by Lawrence, Simmons and Wotton [5, 35].

## Place recognition

The results shown in Fig 5 indicate that the encoded cochleograms also retained sufficient information to allow a neural network to classify each echo as belonging to the location at which it was collected. The network did experience some difficulties assigning a few encoded cochleograms to their respective locations. Analysis of the confusion matrix reveals that the network mostly confuses somewhat similar locations. For example, echoes collected in one room of a private residence were assigned to another room. Also, echoes from one field site were classified as another field site.

## Discussion

From cochleograms of ensonification data collected in various bat habitats and indoor environments, we derived an efficient encoding based on 25 independent components. These independent components can be conceptualized as neurons with particular spectrotemporal receptive fields whose output yields a compressed description of the cochleograms [14]. Applying this encoding to a cochleogram is not lossless, as was shown in Fig 2. Reconstructing a cochleogram from its encoded representation results in a smoothed version of the original. However, this loss of information does not impede performance on the tasks modeled in this paper. Despite being highly compressive, the encoding retains essential information to support

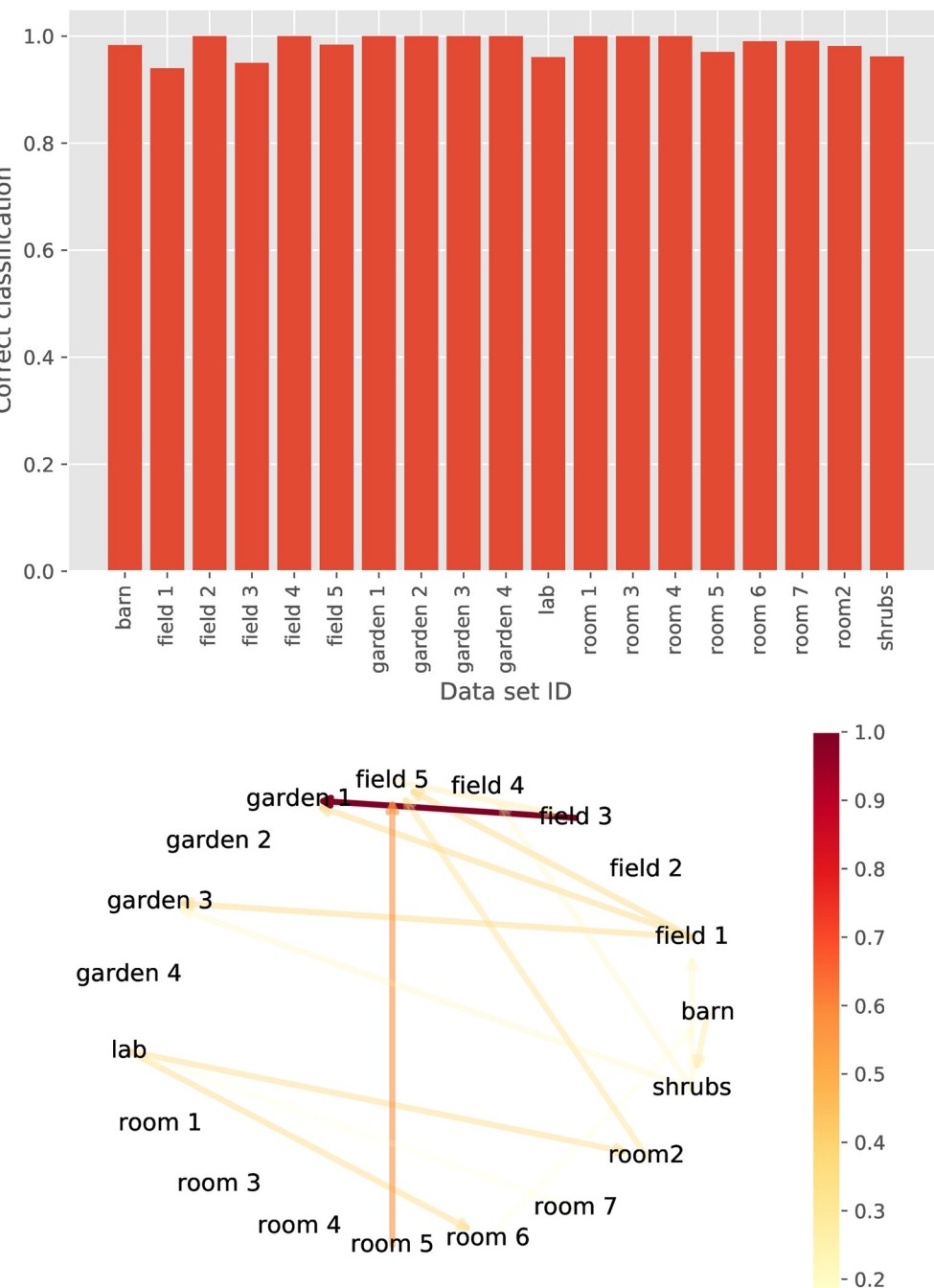

**Fig 5. Performance of the neural network memorizing the class membership (=location) of each encoded cochleogram.** The bottom panel visualizes the confusion between classes. The arrows show how often members of one class were erroneously attributed to another class. The color and brightness of the error reflect the number of errors. From this graph, it can be seen, for example, that members of *Field 3* were most commonly mistakenly classified as *Garden 1*.

several critical sonar-based tasks: precise ranging, spectral discrimination, target discrimination, and scene recognition.

A similar conclusion, i.e., echoic information can be highly compressed while still allowing for good task performance, was reached by several other studies. One bat echolocation study [17] describes a PCA-based compression of the spectral information contained in the emission and the external ear transfer functions of *Eptesicus fuscus*. Following the same approach as here, the compressed encoding of monaural echo information (a mapping onto 8 PCA components) was used as input into an artificial neural network that estimated azimuth and elevation of the echo direction. The authors conclude that the vertical acuity reached by the network was close to that of bats and mostly limited by the coarsely sampled emission and hearing patterns used to derive the PCA encoding. Also, as already mentioned in the introduction, several robotic sonar studies similarly concluded that significant compression could be achieved without compromising task performance [8, 9, 41, 42].

The main difference between this study and these previous studies is that the set of filters used in the proposed compressive encoding is not custom-built for one particular task but is derived from a database of ecologically valid echo signals. This approach reasons that a sonar system, biological and artificial, can profit from learning the ensemble's statistical structure of echo signals it receives from its environment as this will allow it to represent this structure with optimal efficiency. As shown by the results above, a sufficiently accurate representation of this structure is the only requirement for successfully completing the same tasks that can be performed using the information contained in the raw received signals. Our finding that a limited set of filters can efficiently and effectively encode echoic information has implications for the efficiency of information processing and suggests a potential efficient neural implementation for sonar processing. Below, we discuss both implications of the current results.

## Information processing implications

To estimate the data reduction rate accomplished by the encoding, we estimate the number of bits required to encode a cochleogram and compare it with the number of bits required to encode the 25 filters' outputs. In this study, a cochleogram consists of 140000 floating-point numbers (20 frequency channels × 7000 samples). However, the low pass filter in the model of Wiegrebe [11] allows reducing the effective sample rate needed to represent the cochleograms. Assuming that the 1 kHz low-pass filter in the cochlear model is an ideal low-pass filter, completely removing all frequency components above 1 kHz, only 2 kSamples/sec would be required to fully encode the information in each frequency channel of the cochleogram. Hence, at 360 kSamples/sec, the cochleograms are oversampled by a factor of 180. Note that the low-pass filter in the model is not ideal and does not remove all frequency content above 1 kHz. Therefore, in practice, sampling at a somewhat higher Nyquist rate is required.

Next, we need to determine how many bits are required to encode the samples of a cochleogram. Shannon's source coding theorem specifies that a cochleogram containing a large number $N$ of independent and identically distributed samples with each sample $S(t_j, f_k)$ having an entropy $H(S) = -\sum_i p_i * \log_2(p_i)$ can be compressed into $N \cdot H(S)$ bits with negligible risk of information loss [43]. From the empirically derived distribution $p_i$ of the cochleogram values (Fig D in S1 Text) we estimated $H(S) = 4.40$ bits. As this value depends on the quantization used, we chose to encode the sample values as doubles. The information capacity $I_S$ of a cochleogram would then be about 3422 bits,

$$I_S = \frac{140000 \text{ samples} \times 4.40 \text{ bits per sample}}{180 \text{ oversample factor}} \tag{5}$$

However, not all cochleogram samples are identically distributed, as can be seen from Fig C in S1 Text showing the average cochleogram derived from the database. Indeed many samples contain hardly any energy (and, thus, information). As a first and rough approximation to the real value of $I_S$ we limited the samples to be encoded to those belonging to the region of the average cochleogram where the values attained at least 20% of the maximal value (see also Fig C in S1 Text). On average, this retained 114,212 samples per cochleogram, resulting in a more accurate estimate of $I_S$ = 2792 bits

$$I_S = \frac{114,212 \text{ samples} \times 4.40 \text{ bits per sample}}{180 \text{ oversample factor}} \tag{6}$$

This estimate assumes temporal and spectral independence between samples of the cochleogram. However, nearby samples of the cochleograms are highly correlated. Removing these redundancies allows for more efficient encoding. Indeed, in this study, we showed that the cochleograms could be encoded using only 25 independent and identically distributed coefficients without compromising performance in a broad range of echolocation tasks. Using the same quantization used for the cochleogram samples, we can again calculate the entropy of each of those coefficients $H(C)$ based on the empirically derived distribution of their values (see Fig D in S1 Text). With $H(C)$ = 13.29 bits this results in an information capacity $I_C$ of the compressed encoding given by $I_C$ = 25 × 13.29 = 332 bits.

While several approximations were introduced to derive these values, they indicate that the cochlear output's informational load can be reduced by roughly 90% through compressive encoding. Despite this large reduction in information content, the simulations reported above show that this compressed representation is sufficient to solve a broad range of sonar-based discrimination tasks. The possibility of reducing the informational load while retaining essential information should allow highly efficient processing, memorization, and retrieving of sonar-based percepts in bats. Because processing more (complex) information is expensive, both in terms of metabolism and the required neural substrate [2, 44], the ability to extract the relevant information also results in decreased costs to the animal.

## Physiological implications

Our modeling results show that complex and simple echoes can be effectively represented as the sum of a few components (filters). This was demonstrated by showing that the components retain sufficient information to address several echolocating tasks. This indicates that, at least in principle, the bat could encode the echoic information present at the cochlear nucleus level (modeled here by the cochleograms) using a set of similar components at some higher up stage in the auditory pathway. And use the result to address the echolocation tasks we model.

Evidence from other sensory domains shows that filters are a common way to encode sensory input efficiently. Most famously, so-called simple cells in the primary visual cortex of mammals have been described as applying filters to an image, the output of which determines its response (spike rate) [45]. The receptive fields of these cells have been likened to Gabor filters [46] having optimal localization in both the spatial and the spatial-frequency domains [47], and, therefore, provide an efficient edge encoding scheme [46].

Filters for efficient encoding of sensory information have not only been found in the visual pathway but also the auditory pathway of several mammalian species. In particular, Andoni et al. [48] reported on inferior colliculus cells in the bat *Tadarida brasiliensis* whose receptive fields have some similarity to the filters we derived. The cells' responses to communication calls could be predicted from filters with similar spectrotemporal receptive fields, or non-linear

combinations thereof [49]. In other mammals as well, neurons with similar spectrotemporal receptive fields have been found (See [50] for references).

If a set of filters can be used for efficient processing of echoic information, could these filters be implemented neurophysiologically? An extreme interpretation of the current results would be that bats require only 25 neurons (or a similarly small number) with the right spectrotemporal responses to echolocate successfully. A more biologically plausible suggestion is that the bat approximates this sparse coding of echoic information by having populations of neurons that, *as an ensemble*, extract components from the echoic input. The pooled output of these ensembles could be used by other centers to support decision-making or flight control. In this view, motor control or decision-making could be based on reading out 25 (or a similarly small number) of auditory pathway populations.

Assuming that the filters could be implemented as populations of cells allows for a lot of freedom in the response properties of the individual neurons in each ensemble. For example, [48] performed a similar analysis on social calls of bats than we did on sonar data. These authors derived a set of theoretical filters. They found that some individual cells responded as predicted by the theoretical filters. However, in their follow-up paper [49], they found other cells to act as if they implemented non-linear combinations of the theoretical filters. This indicates that at least in the encoding of social calls, the auditory system exploits the input signals' mathematical properties, decomposing them into independent components, which is an effective encoding strategy. However, their results also indicate that this decomposition is less straightforward than the theoretical filters derived through Independent Component Analysis or similar techniques.

The receptive fields of the neurons observed by Andoni et al. [48] and others do not encode delay or distance information as these aspects are irrelevant for interpreting communication sounds. Instead, they capture those spectrotemporal features relevant to represent the ensemble of communication calls. In contrast, the filters we derive here encode the time-of-arrival (distance) of the echoes and their spectral content. As such, we consider the proposed filters to be analogous to those observed in *Tadarida brasiliensis* (and other species) but optimized for encoding sonar information by representing the primary cues for a sonar system: delay and spectral information. While such filters have not been directly observed, their required predecessors exist in echolocating bats' auditory pathway.

Frequency selective cells are prevalent throughout the auditory pathway of bats, which is largely tonotopically organized [51]. Poon et al. [52] reported the inferior colliculus of the FM-bat *Eptesicus fuscus* to be tonotopically organized. See also [53, 54] for frequency tuned IC cells in *Eptesicus fuscus*. Frequency selective cells are present up to the cortex, where tonotopically organized areas have been found in several species (See Kossl et al. [55] for references). Target-distance selective cells have been found in the inferior colliculus of *Pteronotus parnellii*, a CF-FM bat [56] as well as in the inferior colliculus of an FM bat *Eptesicus fuscus* [54]. The auditory cortex of several CF-FM bats contains echo distance maps [55]. Such organized maps have been also found in some species of FM bats, *Carollia perspicillata* and *Phyllostomus discolor* [57, 58] but not in others [59–61], see Kossl et al. [62] for a review. Interestingly, target-distance selective neurons in both CF-FM and FM bats show broader delay-tuning when being selective to longer target-distances [63]. This feature is also present in the basic components reported in the present paper, which show increasing response time (i.e., respond to a broader range of target-distances) with increasing distance (see Fig 4E–4H).

Cells with selective delay and frequency responses are sufficient predecessors for establishing populations of cells with spectrotemporal receptive fields mimicking those of the filters we propose here. Moreover, as both frequency selectivity (already present on the level of the cochlea) and delay selectivity co-exist from the IC upwards, the filters could be established at various processing stages. This would also be in line with neural processing concepts along the

ascending auditory pathway [64]. According to these concepts, auditory feature extraction occurs on lower stages of the auditory pathway up to the inferior colliculus. In the auditory cortex, these features are then organized into auditory objects or sensory maps. However, it should be mentioned here that top-down feedback could influence feature extraction and spectro-temporal filter properties. Typically, frontal cortical areas are thought to be involved in top-down control of sensory processing [65–68] and decision making [69].

## Conclusion

We have shown that complex echoes from real environments can be efficiently and effectively represented using a small set of filters. The redundancy in echoic information opens up the opportunity for efficient encoding, reducing the computational load of echo processing and the memory load for storing the information. Therefore, we predict the auditory system of bats to capitalize on this opportunity for efficient coding by implementing filters with spectro-temporal properties akin to those hypothesized here.

## Supporting information

**S1 Text. Fig A**. Results from the procedure used to determine a realistic level of internal noise also used in references [7, 11, 27]. As described in the main text, for each level of noise $\sigma$ we determined $P[\sum C_6 + \mathcal{N}_\sigma > \sum C_0 + \mathcal{N}_\sigma]$, using a Monte Carlo approach. This graph gives the probability $P[\cdot]$ as a function of noise level $\sigma$. The value of $\sigma$ at which $P[\cdot] = 0.75$ was taken as the noise level throughout this paper. **Fig B**. Ensonification device. The custom-built device consisted of two Knowles microphones embedded in a 3D printed housing and a Sensecomp 7000 emitter. The device was mounted on a tripod. **Fig C**. Average cochleogram, derived from the ensonification data. b Binarized average cochleogram showing where samples are at least 20% of the maximum value. **Fig D**. (Left) Distribution of the values of the cochleograms collected in this paper. (Right) Distribution of the 25 filter output values for all ensonification data echoes used in the this paper. As the value of the entropy H(S) depends on the quantization used, we encoded all samples as doubles for this calculation.
(PDF)

## Author Contributions

**Conceptualization:** Adarsh Chitradurga Achutha, Herbert Peremans, Dieter Vanderelst.

**Data curation:** Dieter Vanderelst.

**Formal analysis:** Adarsh Chitradurga Achutha, Uwe Firzlaff, Dieter Vanderelst.

**Investigation:** Dieter Vanderelst.

**Methodology:** Adarsh Chitradurga Achutha, Herbert Peremans, Dieter Vanderelst.

**Supervision:** Dieter Vanderelst.

**Visualization:** Dieter Vanderelst.

**Writing – original draft:** Dieter Vanderelst.

**Writing – review & editing:** Herbert Peremans, Uwe Firzlaff, Dieter Vanderelst.

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
