## [Decision Letter · Decision Letter 0]

11 Feb 2021

Dear Dr. Vanderelst,

Thank you very much for submitting your manuscript "Efficient encoding of spectrotemporal information for bat echolocation" for consideration at PLOS Computational Biology.

As with all papers reviewed by the journal, your manuscript was reviewed by members of the editorial board and by several independent reviewers. In light of the reviews (below this email), we would like to invite the resubmission of a significantly-revised version that takes into account the reviewers' comments.

We cannot make any decision about publication until we have seen the revised manuscript and your response to the reviewers' comments. Your revised manuscript is also likely to be sent to reviewers for further evaluation.

Sincerely,

Emma K Towlson

Guest Editor

PLOS Computational Biology

Natalia Komarova

Deputy Editor

PLOS Computational Biology

Reviewer's Responses to Questions

**Comments to the Authors:**

Reviewer #1: The paper by Achutah et al. titled: Efficient encoding of spectrotemporal information for bat echolocation analyzes how much information is contained in complex real-world echoes. The work is novel and important revealing how a low dimensionality version of an echo can maintain the majority of information contained in the original echoes. This is highly relevant for the central nervous system that has to represent the high dimensionality world in much lower dimensions. The authors examine four fundamental tasks in echolocation – temporal accuracy, resolution, IR classification and localization (elevation) showing that the low representation echoes can be used to explain bats’ performance in these tasks. The work is thus very comprehensive. In general, I found the paper well written and I had rather few comments.

One criticism I have is that the authors could use the neural representation of the inner each which already reduces the high dimension correlogram to a lower dimension in a way that is more biological than an ICA analysis. I know of no evidence that the brain transforms sensory information into independent axes – in fact this was the view in the vision literature ca. 2 centuries ago, but is not generally accepted today. This criticism is related to the authors statement that: “The main difference between this study and these previous studies is that the set of fllters used in the proposed compressive encoding is not custom-built for one particular task but is derived from a database of ecologically valid echo signals.”

From my understanding, the authors used ICA which has some compressive definitions, but one could say the same thing about PCA which has been used before. I thus do not understand why this statement is correct.

My second main major concern is the use of neural networks and their tendency to overfit the data, even when training and testing sets are supposedly separated. For example, in many studies samples are divided randomly between training and testing sets, even though samples are often highly correlated, e.g., when two echoes are collected from nearly the same angle – they will be very similar. In such cases, there is essentially no separation between the training and the testing sets. From reading the manuscript, it seems to me that a similar approach was taken here. If this is correct, the performance of the network shows an ability to learn a set of examples rather than to generalize. To test this, or prove it wrong, the authors should measure similarity between examples in the training and testing sets and show that there are no or few examples which are very similar.

Minor comments:

L119 – The authors write: “We averaged across these three repeats to increase the signal to noise ratio”

Was the averaging done at the correlogram level? Averaging at the level of the time-signal would change the signal dramatically

I think that the notation of the math (starting around line 135) is confusing. In theory the ICA could have run directly on the correlograms, but from what I understood the authors first use PCA to reduce correlogram dimensionality to 25 and only then ran an ICA analysis. The application of a PCA in the middle is not shown in figure 1, so it is not fully clear to me if and when it was performed. Moreover, the authors use S (capital or not) to notate three different things. I think this could improve much

Note that Yovel et al. 2008 already showed that a much lower dimension representation of the echo can be used to perform object classification. If I am not mistaking, they used PCA to reduce the data to ca. 200 dimensions, essentially using a similar approach to that described here.

Figures are not numbered according to their appearance in the paper

Reviewer #2: The paper by Achutha et al. investigates how an efficient encoding of sensory representations could support echolocation tasks in bats. The authors obtain cochleograms from and ensemble of echo signals collected in various outdoor and indoor environments, and use these representations to derive a low-dimensional, efficient encoding of the spectrotemporal characteristics of the outputs of the cochlear models. In order to show that such compressive encoding retains sufficient information to support high echolocation performance in bats, neural networks were trained in tasks which mimicked previous experimental approaches using real animals. The experiments conducted in this study show that neural networks can exploit the information contained in the low-dimensional representation of echoes to solve two-alternative-force-choice tasks requiring high temporal and spectral resolutions. The authors conclude that the efficient encoding of echo information is feasible, and that it could likely find neural correlates in the bat’s auditory system.

Previous studies have demonstrated that low-dimensional encodings allow for task resolution at a relatively high level. One of the advantages of the approach taken here is that the authors were able to show that high performance can also be achieved when low-dimensional filters are derived from a dataset of realistic echo signals, not tailored to specific experimental conditions. The manuscript is well-written, and the conclusions well supported. However, there are a few things that require clarification and amendment before publication. My specific comments are below:

Introduction

• The first citation in the paper is [55]. What’s happening is that references are listed alphabetically in the Reference list first, and only then referenced in the main text. This can be confusing at times for the reader who is trying to check the citations.

• Line 17-18: the sentence is unclear. Probably related to the use of the word “foremost” in this context? For example, do the authors mean that the timing of echo arrival is the most relevant information?

• Line 29: “Pose” recognition by Vanderelst et al [52]?

Methods

• Line 105: “just over 1000 echoes”. Is it above 1000 echoes, as this suggests, or were there N=1000 echoes, as stated above (l. 70)?

• Line 108: This part is unclear to me. Echoes were 23 ms long. Were all echoes equally long, regardless on the reflector (21 locations, etc.), or was echo length defined as 23 ms for convenience? It is also unclear how the maximal range is estimated from the echo duration; if this is derived by the authors, then it should be explained in the Methods. If it’s based on previous literature (e.g. [49]), it should be specified.

• Line 127: hyperbolic FM sweeps are maximally Doppler-shift resilient. Can the authors provide a citation for this?

• As far as I understand, the de-chirping requires to know a reference signal (the way is done, for example for radar detection). The authors claim that the “bat has knowledge of its emission”, yet no reference is provided for this, nor a mechanism suggested (e.g. efference copies?). Other models that compensate the FM effect (like that of Wiegrebe, 2008) base this compensation on computing autocorrelations. Autocorrelations could be calculated from neural activity given, for example, a plausible architecture such as described by Shamma (2001, Trends in Cognitive Sciences). Such architecture requires no knowledge of a “reference” signal, like the de-chirping does. Why was the de-chirping implemented in the model? Would the FM compensation via autocorrelation produce the same results? Could the authors substantiate the phrase “the bat has knowledge of its emission” with a speculation of how this “knowledge” occurs in the bat’s brain?

• Line 208: JND is not defined as an abbreviation.

• Line 221: where instead of were.

Results

• Fig. 3e-h are mentioned first in the main text. Why not swap panels a-d (top) with e-h (bottom)? This way, the order would be consistent in the main text, and also individual component examples would be shown first, before exploring details about components in general.

• Fig. 3a: x-axis and label missing?

• Fig. 3d: although distance and time and linearly dependent, this relationship is not explicit in the manuscript. The authors write in the main text (l. 360) of “best delay”, when in the figure is shown “Filter best distance (m)”. I suggest to either make the relationship [ms] -> [m] explicit, or modify Fig. 3d to show best delays and not best distance for consistency.

• Lines 386-387: JND is not defined in the text, it would be better to define it and to specify the parameter you are freeing to, i.e. sometimes JND is in ms, sometimes in dB.

Discussion

• As enjoyable as the Information processing implications subsection is, I have my reserves as to how it fits into the Discussion. Given its reliance on calculations and data (e.g. Figs. S3, S4), could it be better placed in the Results? It is, at any rate, unusual to bring up new data and figures for the first time in the Discussion section.

• Line 594-96: The authors say, based on ref. 20, that bat’s neurons show broader (frequency) tuning when being selective to longer target distance. Yet, they also say that this feature is reflected in the paper, in which increasing response time occurs with increasing target distance (from Fig. 3e-h, I suppose). Unless I miss something, broader frequency tuning (as per ref. 20) is not necessarily equivalent to increasing response times, although both seem to correlate with large delays. Perhaps it would be useful to clarify this.

• Line 599: frequency selectivity does not appear from IC on but from the cochlea onwards and persist along the pathway.

• Lines 593-95: what centers would support decision making during echolocation in bats? I think the literature is very conflicting about this. Would it involve cortex or only subcortical structures?

• Can the 25 filters reported here suffice also for communication call coding in this species?

• General: The neurophysiology part of the discussion is very “feedforward” focused. Yet it is almost certain that feedback pathways play a role when the bat vocalizes. For example feedback activity could change the way the 25 filters reported here behave in different contexts. This should be pointed out to prevent the naïve reader from getting the impression that everything is feedforward in the bat brain.

Others

• Fig. S4 is missing y-axis labels.

• Also Fig. S4 (right). The figure caption reads that the distribution shown here is for 15 filter output values. Does that mean that there were 15 values? Or that this is the distribution of values from 15 filters? If the latter is correct, and there were 25 filters from the ICA analysis, is this a typo?

**Have all data underlying the figures and results presented in the manuscript been provided?**

Reviewer #1: Yes

Reviewer #2: None

PLOS authors have the option to publish the peer review history of their article (what does this mean?). If published, this will include your full peer review and any attached files.

Reviewer #1: No

Reviewer #2: **Yes: **Julio Hechavarria
---

## [Decision Letter · Decision Letter 1]

7 May 2021

Dear Dr. Vanderelst,

We are pleased to inform you that your manuscript 'Efficient encoding of spectrotemporal information for bat echolocation' has been provisionally accepted for publication in PLOS Computational Biology.

Best regards,

Natalia L. Komarova

Deputy Editor

PLOS Computational Biology

Natalia Komarova

Deputy Editor

PLOS Computational Biology

Reviewer's Responses to Questions

**Comments to the Authors: **

Reviewer #1: The authors have answered my questions. 

I accept the point regarding performing vs. generalizing, but I think that this should be emphasized in the paper. The authors show that the task can be solved with the compressed data, but they show no generalization which bats are known to possess (even if not shown in these tasks)

Reviewer #2: The authors hace dealt with all my comments and concerns in a propoer way. I think the paper is ready for publication as is. I can't but congratulate the authors on this excellent work.

**Have the authors made all data and (if applicable) computational code underlying the findings in their manuscript fully available?**

Reviewer #1: Yes

Reviewer #2: None

PLOS authors have the option to publish the peer review history of their article (what does this mean?). If published, this will include your full peer review and any attached files.

Reviewer #1: No

Reviewer #2: **Yes: **Julio C. Hechavaria

---

## [Editor Report · Acceptance letter]

21 Jun 2021

PCOMPBIOL-D-20-01765R1 

Efficient encoding of spectrotemporal information for bat echolocation

Dear Dr Vanderelst,

I am pleased to inform you that your manuscript has been formally accepted for publication in PLOS Computational Biology. Your manuscript is now with our production department and you will be notified of the publication date in due course.

With kind regards,

Olena Szabo
